# Structural characterization and dynamics of AdhE ultrastructures from *Clostridium thermocellum* show a containment strategy for toxic intermediates

Samantha J Ziegler[1], Brandon C Knott[1], Josephine N Gruber[1], Neal N Hengge[1], Qi Xu[1], Daniel G Olson[2], Eduardo E Romero[3†], Lydia-Marie Joubert[4], Yannick J Bomble[1]*

[1]Biosciences Center, National Renewable Energy Laboratory, Golden, United States; [2]Thayer School of Engineering at Dartmouth College, Hanover, United States; [3]Department of Biochemistry and Molecular Genetics, University of Colorado Anschutz Medical Campus, Aurora, United States; [4]SLAC National Accelerator Laboratory, Menlo Park, United States

**\*For correspondence:**
Yannick.Bomble@nrel.gov

**Present address:** [†]University of Nebraska-Lincoln, Lincoln, United States

**Competing interest:** The authors declare that no competing interests exist.

## eLife assessment

This work presents **valuable** information on the structure of the spirosome's native extended conformation as the active form of the aldehyde-alcohol dehydrogenase (AdhE) enzyme. The evidence is **solid**, although the work does not provide a mechanistic understanding of the function and dynamics of AdhE.

**Abstract** *Clostridium thermocellum*, a cellulolytic thermophilic anaerobe, is considered by many to be a prime candidate for the realization of consolidated bioprocessing (CBP) and is known as an industry standard for biofuel production. *C. thermocellum* is among the best biomass degraders identified to date in nature and produces ethanol as one of its main products. Many studies have helped increase ethanol titers in this microbe; however, ethanol production using *C. thermocellum* is still not economically viable. Therefore, a better understanding of its ethanol synthesis pathway is required. The main pathway for ethanol production in *C. thermocellum* involves the bifunctional aldehyde-alcohol dehydrogenase (AdhE). To better understand the function of the *C. thermocellum* AdhE, we used cryo-electron microscopy (cryo-EM) to obtain a 3.28 Å structure of the AdhE complex. This high-resolution structure, in combination with molecular dynamics simulations, provides insight into the substrate channeling of the toxic intermediate acetaldehyde, indicates the potential role of *C. thermocellum* AdhE to regulate activity and cofactor pools, and establishes a basis for future engineering studies. The containment strategy found in this enzyme offers a template that could be replicated in other systems where toxic intermediates need to be sequestered to increase the production of valuable biochemicals.

## Introduction

Lignocellulosic biomass is one of the most attractive substrates for sustainable production of second-generation biofuels and other bioproducts (*Ashokkumar et al., 2022*; *Biddy et al., 2016*; *Demain, 2009*; *Isikgor and Becer, 2015*; *Usmani et al., 2020*). Consolidated bioprocessing (CBP) is emerging

as a promising process to reduce costs by combining biomass solubilization and fermentation in one step without added enzymes (*Fan, 2014*; *Lynd et al., 2002*; *Lynd et al., 1999*; *Lynd et al., 2005*). Due to its ability to quickly solubilize and utilize cellulose, *Clostridium thermocellum* is an ideal candidate organism for CBP and has previously been engineered to produce ethanol at high yield (*Akinosho et al., 2014*; *Verbeke et al., 2017*). However, ethanol production using *C. thermocellum* is not yet an economical process due to limited titers (*Brown et al., 2011*; *Rani and Seenayya, 1999*; *Tian et al., 2019*). The main pathway for ethanol production in *C. thermocellum* involves the bifunctional aldehyde-alcohol dehydrogenase (AdhE), a gene that is often modified in strains selected for alcohol tolerance and other selective pressures, indicating its crucial role in *C. thermocellum* metabolism (*Tian et al., 2019*; *Lo et al., 2015*; *Olson et al., 2023*; *Shao et al., 2011*). AdhE functions in the anaerobic fermentation pathway and contains two domains: an aldehyde dehydrogenase (ALDH) domain for the reduction of acetyl-CoA to acetaldehyde and an alcohol dehydrogenase (ADH) domain to reduce acetaldehyde to ethanol. Both reduction processes natively use NADH as a cofactor and are reversible in the presence of $NAD^+$ as a cofactor. Because AdhE is part of the native ethanol production pathway in many anaerobic bacteria, it is a key target protein in many biofuels and biochemicals-related studies (*Tian et al., 2019*; *Lo et al., 2015*; *Shao et al., 2011*; *Loder et al., 2015*; *Zheng et al., 2015*; *Biswas et al., 2015*).

Structurally, AdhE is an intriguing protein as it forms large, filamentous helical ultrastructures termed spirosomes, that could be necessary for function (*Kessler et al., 1992*; *Matayoshi and Oda, 1985*). In the case of *Echerichia coli* AdhE, spirosome formation seems vital for the forward reaction of acetyl-CoA to ethanol, while the reverse reaction is not impacted by the disruption of the ultra-structures (*Kim et al., 2019*; *Pony et al., 2020*; *Espinosa et al., 2001*). Prior structural studies on the *E. coli* AdhE complex captured both the extended and compact forms of the spirosome, with some biochemical evidence that the extended form was required for the reduction of acetyl-CoA to ethanol (*Kim et al., 2019*; *Pony et al., 2020*; *Kim et al., 2020*). Because *C. thermocellum* AdhE and *E. coli* AdhE have 62% sequence identity, we first aimed to determine the similarity between the AdhE from *C. thermocellum* and *E. coli*. To further understand the AdhE from *C. thermocellum*, we used cryo-electron microscopy (cryo-EM) to capture the structure of the spirosome ultrastructure in both its extended and compact forms. We further analyzed the results of our 3.28 Å extended *C. thermocellum* AdhE in combination with channel prediction software and molecular dynamics (MD) simulations. We determined that the native state of the *C. thermocellum* spirosome is extended (different from that of other structurally characterized AdhEs) and contains an enclosed channel between the ALDH and ADH active sites. In modeling the compact spirosome, we found that this channel leaks acetaldehyde to the bulk phase, which indicates that the extended spirosome might be crucial for toxic aldehyde intermediate channeling.

## Results and discussion

### Spirosome conformation is determined by intrinsic sequence and local environment, but not by expression host

A fundamental aspect of this work was determining if the local environment of AdhE affected the formation of the spirosome and its conformation (extended vs compact). This was especially important given that our structural studies utilized *C. thermocellum* AdhE heterologously expressed and purified from *E. coli*. Based on negative stain transmission electron microscopy (TEM) data shown in *Figure 1A*, we found that spirosomes maintained the same conformation between endogenous and exogenous expression – both primarily extended in the case of *C. thermocellum*, which represents the majority *apo* conformation of the *C. thermocellum* spirosome. Intriguingly, the extended conformation of the *C. thermocellum* spirosome remained unchanged when purified from *E. coli*, even though the native *apo E. coli* AdhE is purified primarily in the compact form (*Figure 1B*). Since Kessler et al. found that the addition of both $NAD^+$ and $FeSO_4$ was necessary to transform *E. coli* spirosomes from their compact to their extended forms, we investigated the effects of different reactants that could contribute to the inverse process in the purified, extended *C. thermocellum* spirosome (*Kessler et al., 1992*). While our results were not as extreme as those of Kessler et al., we found that the addition of the forward reactants (NADH, acetyl-CoA, $FeSO_4$) led to the compaction of spirosomes. Conversely, the addition of the reverse reactants ($NAD^+$, ethanol, CoA, $FeSO_4$) to the compact *E. coli* spirosomes

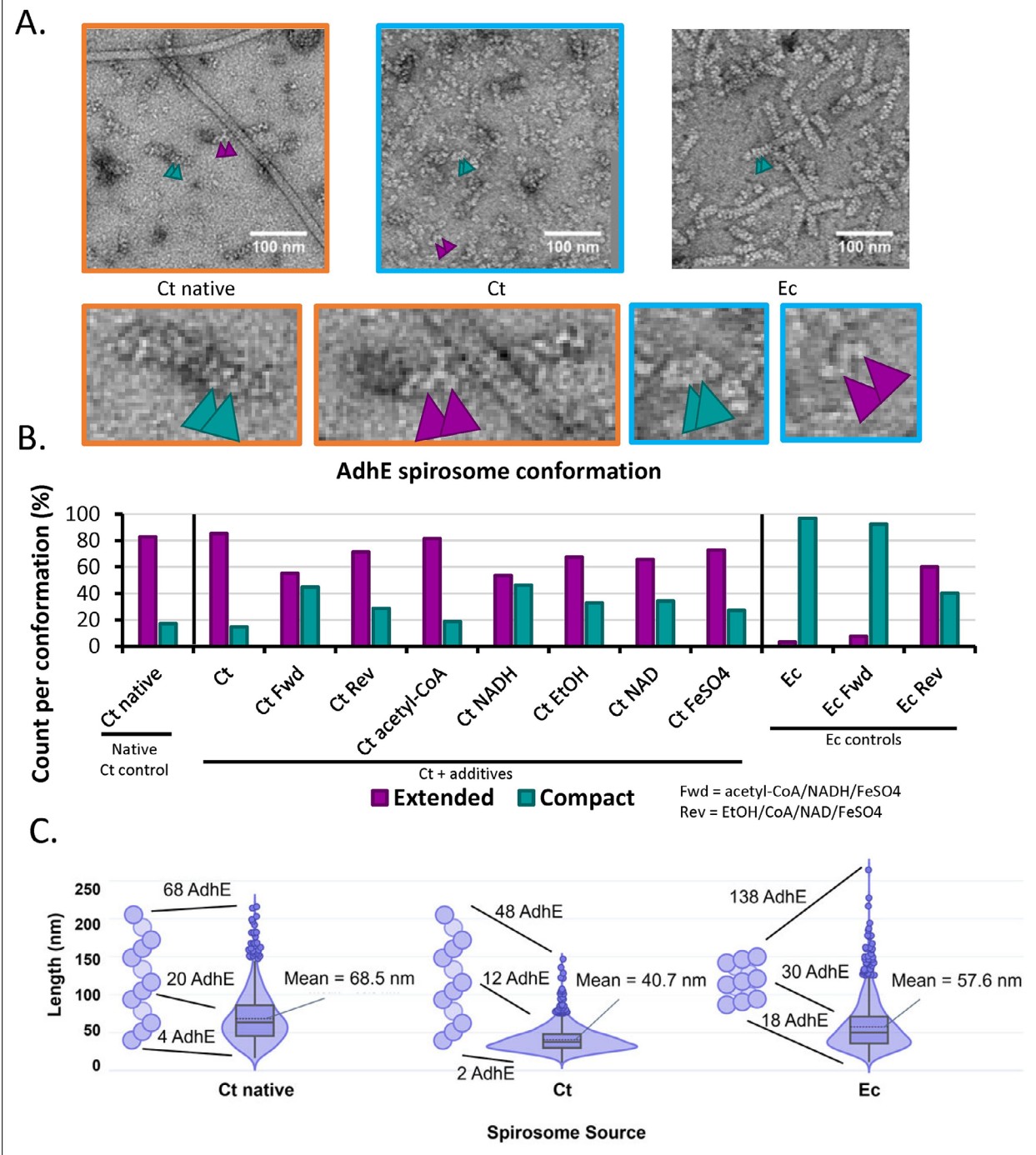

**Figure 1.** Negative-stain data of aldehyde-alcohol dehydrogenase (AdhE) spirosomes indicate that conformation differs between bacteria. (**A**) Sample negative stained images. Ct = *C. thermocellum*, Ec = *E. coli*. Compact spirosome indicated by teal arrows, extended by purple arrows. Zoom panels correlate by color. (**B**) Measurement of spirosome conformation as determined in at least 500 instances. (**C**) Violin plot of the lengths of spirosomes measured in 1250 instances – example spirosomes to the left of each group represent the majority conformation in the *apo* state. The difference in mean length was statistically significant for all three samples as determined by a Mann-Whitney U test.

The online version of this article includes the following figure supplement(s) for figure 1:

**Figure supplement 1.** Negative stain images of *C. thermocellum* aldehyde-alcohol dehydrogenase (AdhE) in the presence of reactants.

resulted in more extended forms. To isolate which substrate was causing this shift, we made negative stained grids of the *C. thermocellum* spirosomes in the presence of each reactant and found that NADH and $NAD^+$ were the major determining factors in the conformation of the spirosome (*Figure 1B*, *Figure 1—figure supplement 1*). We also found that the expression host does not impact the conformation of the spirosomes but does significantly affect their length (*Figure 1C*). *C. thermocellum* AdhE expressed in *E. coli* formed spirosomes that were, on average, approximately 20 nm shorter than those found in a native *C. thermocellum* lysate. This equates to spirosomes that contain, on average, approximately five to ten fewer AdhE monomers in the extended spirosome. The shorter average length was due largely to the lack of spirosomes that were longer than 100 nm in the exogenous expression (*Figure 1C*). This phenomenon has been observed previously with the exogenous expression of *Vibrio cholerae* spirosomes in *E. coli*, which also resulted in shorter spirosomes (*Cho et al., 2021*).

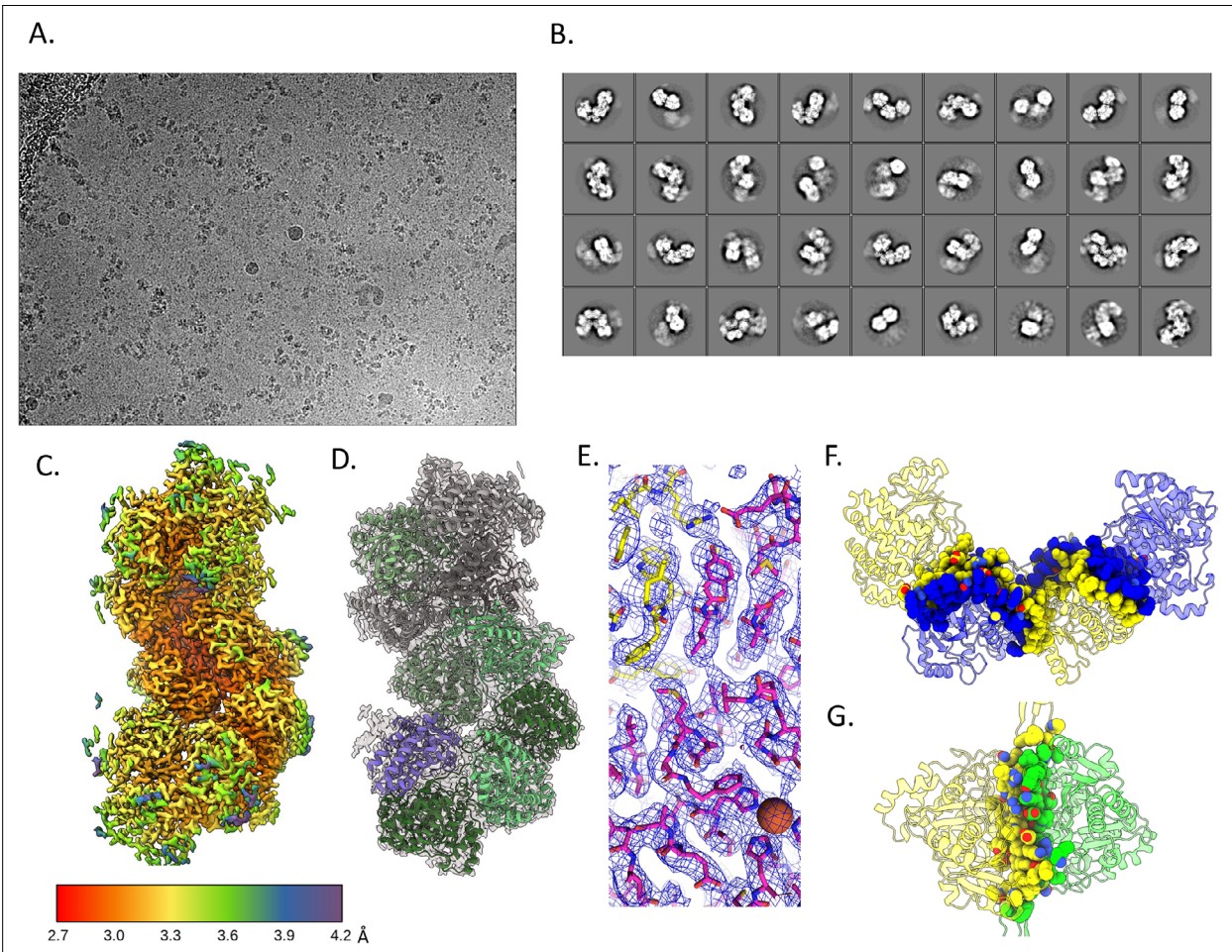

**Figure 2.** Cryo-electron microscopy (cryo-EM) structure of the extended *C. thermocellum* aldehyde-alcohol dehydrogenase (AdhE) spirosome. (**A**) Representative cryo-EM image. (**B**) 2D class averages of the single-particle analysis. (**C**) Local resolution of the final map. (**D**) Cartoon model fit in the density. (**E**) Stick representation to show that side chains can be determined at this resolution, the sphere represents the catalytic Fe in the alcohol dehydrogenase (ADH) domain. (**F**) Dimer interface shows a swapped domain dimerization with a large buried surface area. (**G**) Tetramer interface has a smaller buried surface area.

The online version of this article includes the following figure supplement(s) for figure 2:

**Figure supplement 1.** Comparing structural properties of the interfaces of the *C. thermocellum* and *E. coli* aldehyde-alcohol dehydrogenase (AdhE) spirosomes.

**Figure supplement 2.** Lower-resolution *C. thermocellum* aldehyde-alcohol dehydrogenase (AdhE) structure.

**Figure supplement 3.** Data processing of the *C. thermocellum* aldehyde-alcohol dehydrogenase (AdhE) structure.

## High-resolution structure of the extended *C. thermocellum* AdhE spirosome to analyze oligomerization interfaces

To understand why *C. thermocellum* AdhE is found in the extended spirosome form and *E. coli* AdhE in the compact form, we turned to cryo-EM to obtain a high-resolution structure of the *C. thermocellum* protein complex to analyze its interfaces as compared to previously published *E. coli* spirosome structures at the molecular level (*Kim et al., 2019*; *Pony et al., 2020*; *Kim et al., 2020*). Using standard cryo-EM techniques (see Materials and methods), we were able to determine a 3.28 Å cryo-EM structure of the extended *C. thermocellum* AdhE spirosome (*Figure 2*). We compared multiple aspects of the protein complex to the previously published *E. coli* spirosome structures, including the interaction interfaces, predicted catalytic active sites, cofactor binding pockets, and potential channels that may connect the two active sites (*Kim et al., 2019*; *Pony et al., 2020*; *Kim et al., 2020*).

There are two interfaces that form the spirosome (*Figure 2F and G*). The first is the buried area located between two molecules of AdhE, in a domain-swapped dimer interface, where one ALDH domain is sandwiched between the ADH and ALDH domains of the second protein. This area contains a wide array of residues involved in the interaction that spans the entire protein (*Figure 2F*). The second interface is between the dimer of dimers, here referred to as the tetramer interface. This interface has a much smaller interaction area because it only occurs between two ADH domains (*Figure 2G*). Intriguingly, the overall hydrophobicity and electrostatics of these interfaces are similar between the *C. thermocellum* extended, *E. coli* extended, and *E. coli* compact structures (*Figure 2*, *Figure 2—figure supplement 1*). However, the calculated buried surface area differs between the two *E. coli* dimer interfaces, but not the tetramer interface. Both the *E. coli* and *C. thermocellum*

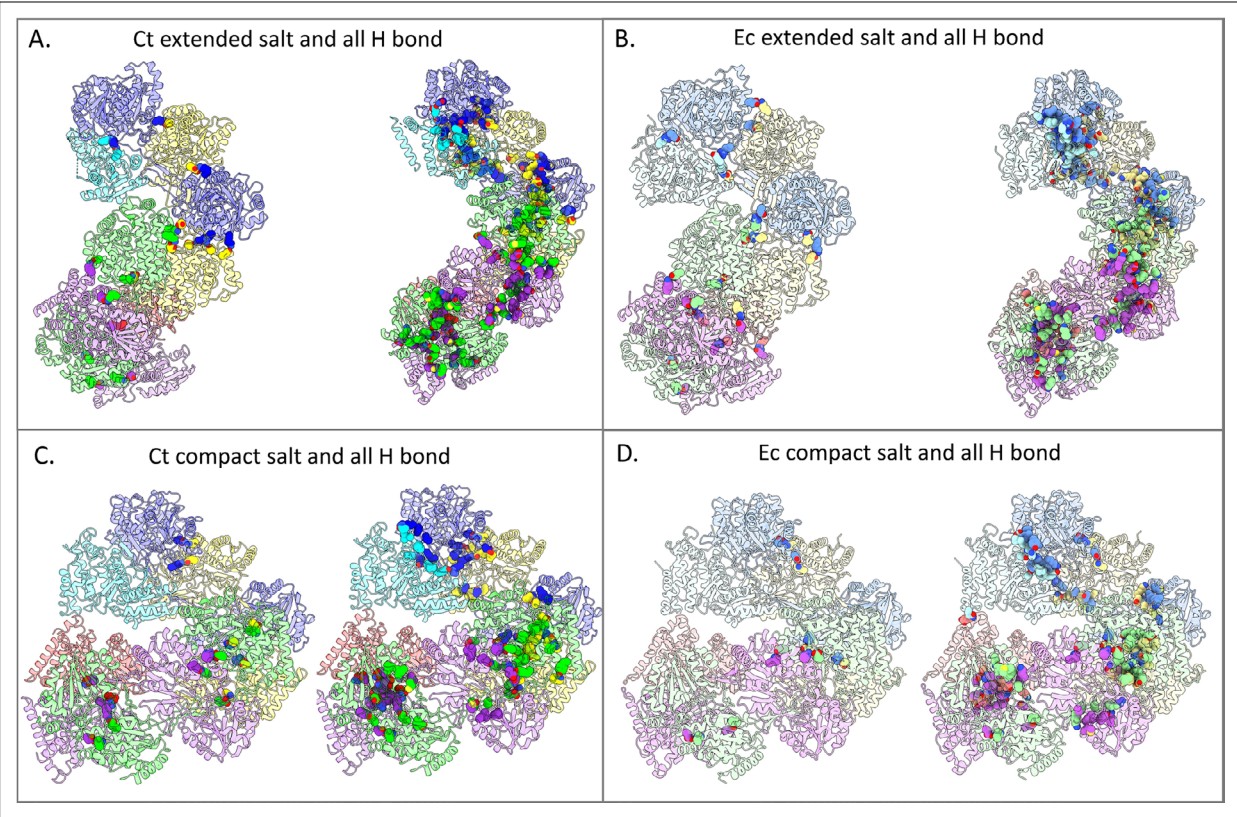

**Figure 3.** Hydrogen bond networks are more prevalent in the extended spirosomes of both *E. coli* and *C. thermocellum*. (**A**) *C. thermocellum* extended spirosome. (**B**) *E. coli* extended spirosome from PDB ID 7BVP. (**C**) SWISS-MODEL of the *C. thermocellum* compact spirosome. (**D**) *E. coli* compact spirosome from PDB ID 6AHC. Individual monomers have unique colors. The *E. coli* structure monomers are colored with lighter shades to correlate to the corresponding monomer in *C. thermocellum*. All salt bridges and hydrogen bonds are represented with spheres to emphasize their locations in the ribbon structure.

The online version of this article includes the following figure supplement(s) for figure 3:

**Figure supplement 1.** Distribution of amino acids found in spirosome interfaces.

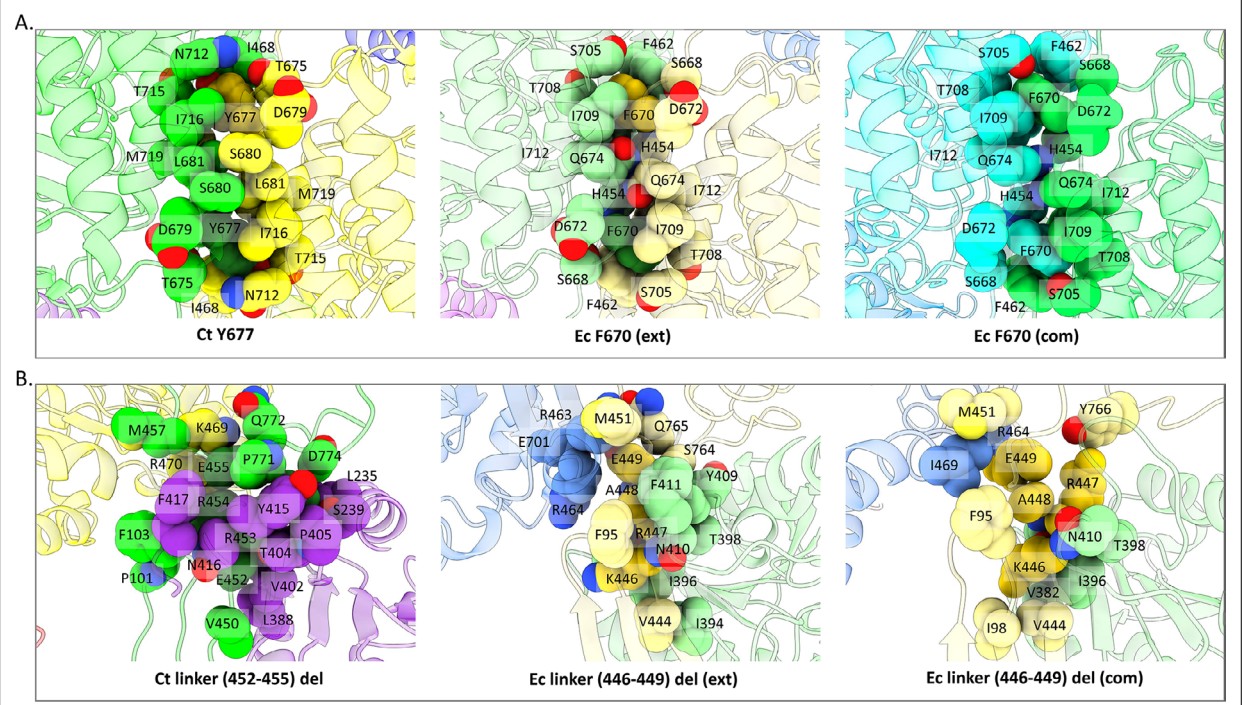

**Figure 4.** Previously identified mutants in *E. coli* compared to the *C. thermocellum* structure. (**A**) Top row represents the mutant *E. coli* F670 that disrupts spirosomes, as well as *E. coli* F670/S705 that lengthens spirosomes as found in PDBID 7BVP. (**B**) Bottom row represents the *E. coli* 446–449 linker deletion as found in PDBID 6TQH. All residues within 5 Å are labeled with their single-letter amino acid code.

The online version of this article includes the following figure supplement(s) for figure 4:

**Figure supplement 1.** Aldehyde-alcohol dehydrogenase (AdhE) active site consensus sequences.

extended dimer interfaces bury ~5000 Å². While the *C. thermocellum* compact dimer interface buries a similar surface area of ~4800 Å², the *E. coli* compact dimer interface buries ~3800 Å². Conversely, all the tetramer interfaces bury ~2000 Å². One would expect the compact structure in *E. coli* to have a larger buried surface area due to it being the predominant form when it is examined without additives, but that is not the case; further corroborating that factors other than buried surface area must impact the *apo* state of the spirosome. Analysis of the residues buried in these interfaces reveals that while many of the residues are identical in the *C. thermocellum* and *E. coli* extended structures, there are some differences in amino acid type distribution, although nothing that directly indicates control of conformer state (*Figure 2*, *Figure 3—figure supplement 1*).

We also examined the salt bridge and hydrogen bond networks between AdhE monomers (*Figure 3*). We found that more salt bridges form in the extended *C. thermocellum* structure as compared to the extended *E. coli* structure, which suggests that this may be a factor in the stability of the extended *C. thermocellum* structure that is not found in the *E. coli* structure. However, when looking between the *E. coli* extended and compact structures, there are two more salt bridges in the extended structure than in the compact structure, which again implies that there might be additional factors impacting the primary form of the spirosome (*Kim et al., 2019*; *Kim et al., 2020*). When looking at the hydrogen bonds between molecules, the extended *C. thermocellum* spirosome has a wider variety of hydrogen bonds (44 unique pairs) than the extended *E. coli* spirosome (32 unique pairs), indicating a potentially more stable structure. However, as with the salt bridges, the compact *E. coli* spirosome has fewer hydrogen bonds (17 unique pairs) than the extended structure, which does not explain why the compact form is the primary structure isolated when purified without substrates.

We also examined areas in the *C. thermocellum* structure that align with areas in the *E. coli* structures that have previously been shown to disrupt spirosome formation or with areas in the *V. cholerae* structure that were implicated in spirosome length. For the former in *E. coli*, there are two examples: (1) F670 mutated to either a small nonpolar amino acid or a charged amino acid (PDBID 7BVP) and (2) the deletion of residues 446–449, both resulting in spirosome dissociation (PDBID 6TQH; *Kim*

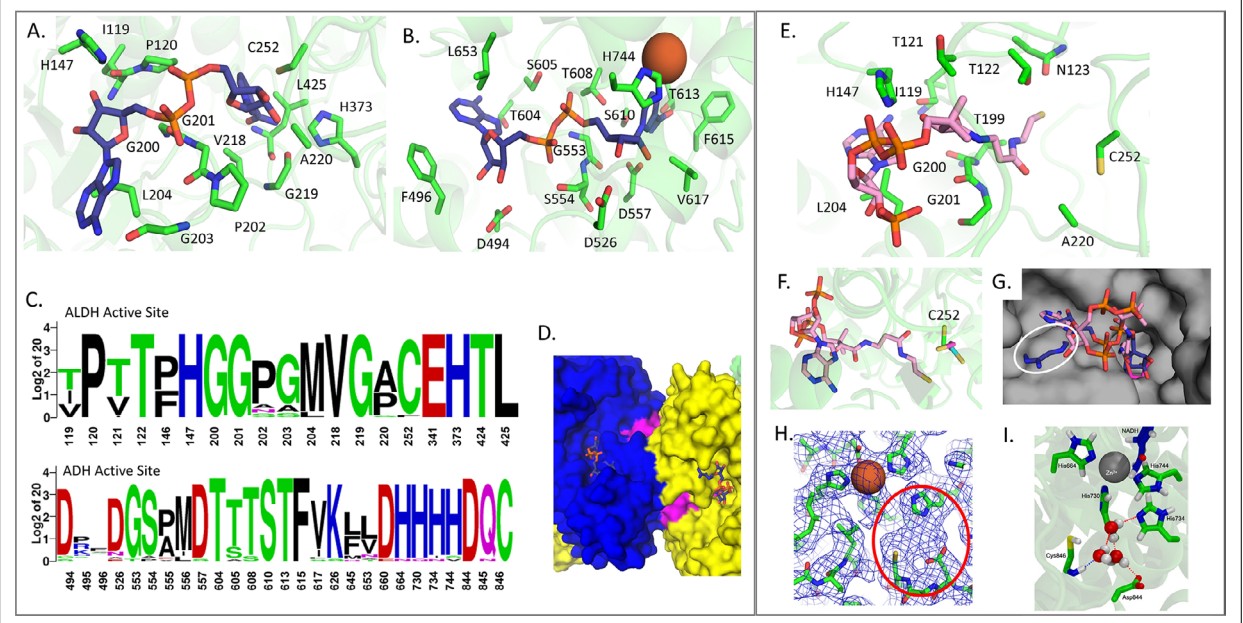

**Figure 5.** Active sites of *C. thermocellum* aldehyde-alcohol dehydrogenase (AdhE) compared to *E. coli* AdhE. (**A**) Aldehyde dehydrogenase (ALDH) NAD⁺ binding domain, where all residues within 2.5 Å of the NAD⁺ are shown in stick (Ct – green, NAD⁺ – navy). (**B**) Alcohol dehydrogenase (ADH) NAD⁺ binding domain, where all residues within 2.5 Å of the NAD⁺ are shown in stick (Fe²⁺ – rust sphere). (**C**) Sequence alignment of the active site residues. (**D**) Surface representation of the novel NAD(P)H binding site (magenta) shown between two AdhE monomers (blue and yellow). (**E**) ALDH binding domain compared to *Rhodopseudomonas palustris* (Rp) bound to acetyl-CoA, where all residues within 2.5 Å of the docked Rp acetyl-CoA are shown as sticks (Ct – green, acetyl-CoA – pink). (**F**) Zoom panel from C showing the catalytic cysteine in relation to the docked acetyl-CoA (Ct – green, Ec – cyan, Rp – magenta). (**G**) Surface view of *C. thermocellum* with both NAD⁺ (Ec, navy) and acetyl-CoA (Rp, pink) docked into the structure; white circle indicates a clash between NAD⁺ and *C. thermocellum*. (**H**) Density of the *C. thermocellum* ADH active site, showing the coordinated Fe²⁺ atom and an empty density between H734, D844, and C846, circled in red. (**I**) Snapshot from molecular dynamics simulation at the ADH active site, illustrating hydrogen bonding between water molecules occupying the active site and the three catalytic residues (H734, D844, and C846, green sticks). Also shown are NADH molecules (navy sticks), H644, H730, and H744 (green sticks), and zinc (gray sphere).

The online version of this article includes the following figure supplement(s) for figure 5:

**Figure supplement 1.** Consensus sequences of the aldehyde-alcohol dehydrogenase (AdhE) channel-lining residues.

---

*et al., 2019*; *Pony et al., 2020*). In *C. thermocellum*, F670 is a tyrosine; however, the hydrophobic pocket that surrounds that residue still exists (*Figure 4A*). We postulate that the similar shape of tyrosine maintains the integrity of this interaction, resulting in stable spirosomes. Regarding the 446–449 deletion, it occurs in the linker between the two domains of the AdhE monomer. This linker is vital for the domain-swapped structure of the dimer. Thus, deleting four residues, regardless of the identity of those amino acids, should interfere with the dimerization of AdhE (*Figure 4B*).

Finally, in *V. cholerae* AdhE expressed in *E. coli* (PDBID 7DAG), Cho et al. found that mutating Y669 and N704 to the corresponding residues in *E. coli*, phenylalanine and serine, resulted in longer spirosomes (i.e. more AdhE subunits in the spirosome) (*Cho et al., 2021*). The *C. thermocellum* sequence aligns with *V. cholerae*, with a tyrosine and asparagine in these positions (*Figure 4A*). However, based on our negative-stain results (*Figure 1C*), we show that *C. thermocellum* AdhE spirosomes, when expressed in *E. coli*, are significantly shorter than when in their native host. We extrapolate from these results that a similar phenomenon could occur if *V. cholerae* spirosomes were isolated from their native organism.

## Structural comparison of AdhE active sites

One factor that compaction and extension of the spirosome affects is the active site of both the ALDH and ADH domains. Here, we compare the NAD⁺-occupied *E. coli* site to the empty site of the *C. thermocellum* structure to determine which residues may interact with NAD⁺ (*Figure 5A and B*). In a sequence comparison between 25 AdhE proteins that have been previously identified in bacteria, both the ALDH and ADH NAD-pockets seem relatively well conserved (*Figure 5C*; *Kessler*

*et al., 1992*; *Matayoshi and Oda, 1985*; *Extance et al., 2013*; *Extance et al., 2016*; *Laurenceau et al., 2015*). There are a few residues with less than 50% identity (when using *C. thermocellum* as the basis of comparison), such as I119, L204, F496, S605, and M645. However, most of these residues retain similar chemistry at those locations – e.g., L204 is a nonpolar residue in every bacterium queried. Conversely, the amino acid that corresponds to the F496 position can vary between nonpolar, negatively charged, and polar, which is surprising since the phenylalanine at that position stacks with the adenosine moiety of the NAD⁺. These patterns generally held true when the sequence alignment was expanded to 1000 AdhE sequences that were identified through the JGI IMG database (*Figure 5*, *Figure 4—figure supplement 1*). Using this larger set of sequences, we also analyzed the D494 position. Prior work has shown that mutating the aspartate to glycine results in a change of cofactor preference from NADH to also include NADPH (*Tian et al., 2019*). There were 2.7% of AdhE sequences that had a naturally occurring G at this position, which suggests that there are AdhE proteins that rely on NADPH for their catalysis. We were able to visualize a potentially novel NAD(P)H binding site previously only identified via sequence comparison (*Figure 5D*; *Zheng et al., 2015*). The sequences contained a conserved GxGxxG motif at residues 426–431, which has been shown to be involved in nucleotide binding (*Bellamacina, 1996*). Intriguingly, this site does not coincide with either the ALD or ALDH active sites and is therefore less likely to have a catalytic function. However, this motif is located at the interface of two AdhE monomers, indicating that it could have an allosteric role, although further studies are needed to confirm. The three glycines at this position are all over 99% conserved, indicating that they play an important role in the AdhE structure or function.

We also used structure alignments to estimate how both the *C. thermocellum* and *E. coli* ALDH active sites would accommodate an acetyl-CoA molecule via comparison to the structure of the *R. palustris* aldehyde dehydrogenase bound to acetyl-CoA (*Figure 5E*; *Zarzycki et al., 2017*). We observed that the proteins aligned with very little deviation – less than 2 Å RMSD – causing the acetyl-CoA aligned from *R. palustris* to sit in most of the same pocket that the NAD⁺ occupies as aligned from *E. coli*. This suggests that the dehydrogenase reaction occurs stepwise to include both the NAD⁺ and the acetyl-CoA (*Figure 5G*; *Zarzycki et al., 2017*; *Lei et al., 2008*; *Rudolph et al., 1968*). Furthermore, when looking at the surface representation of the *C. thermocellum* structure, we found that the NAD⁺ clashes with the protein, indicating that the pocket might shift to accommodate NADH separate from acetyl-CoA (*Figure 5G*). In looking at the residues within 3.5 Å of the acetyl-CoA, we identified multiple regions that seem conserved between all three organisms. Interestingly, if we look at the catalytic cysteine, which is covalently linked to the acetyl-CoA in the *R. palustris* structure, we see that the cysteine for both *C. thermocellum* and *E. coli* are rotated approximately 120° away from the *R. palustris* position in opposite directions, which could be due to the absence of acetyl-CoA in these structures (*Figure 5F*; *Kim et al., 2019*; *Zarzycki et al., 2017*).

At the ADH active site, we observe an empty density in the *C. thermocellum* extended structure, coordinated between residues that have been implicated in the ADH activity of the protein (*Figure 5H*, circled in red) (*Olson et al., 2023*). As illustrated in *Figure 5F*, we see that there is density, most probably a water molecule, being coordinated by H734, D844, and C846. D844 and C846 are 100% conserved, while H734 is 92% conserved among the 25 sequences that we analyzed. While slightly less well conserved in the 1000 sequence comparison, all three residues are still highly conserved (H734 is 97.8%, D844 is 98.8%, and C846 is 91.6% conserved; *Figure 5*, *Figure 4—figure supplement 1*). Furthermore, this small pocket is within 10 Å of the catalytic iron atom, which increases the probability that this region is important for catalysis. No published structure of the ADH domain contains either acetaldehyde or ethanol at the active site, though the structure of the *Geobacillus thermoglucosidasius* ADH domain (PDBID 3ZDR) contains a glycerol molecule coordinating a divalent metal ion (*Extance et al., 2013*). The terminal alcohol group of the glycerol molecule was speculated to be similar to that of ethanol; thus, the structure may constitute a product mimic (*Extance et al., 2013*). Further supporting our hypothesis that a water molecule may occupy the observed empty density at the ADH active site in the *C. thermocellum* extended structure, two independent MD trajectories, each with two full AdhE molecules including two ADH active sites, demonstrate hydrogen bonding of at least one water molecule with D844 in essentially every frame (>98%), as well as with H734 (16–31% of the frames) and with C846 (7–66% of frames with both side chain and main chain, as shown in *Figure 5I*).

# Identification of a channel that confines aldehydes and examination of the dynamics of spirosome ultrastructures

Intermediate channeling has been proposed previously in the *E. coli* spirosome, based on the observation of enclosed space within the structure, and has been hypothesized as a functional role to contain reactive aldehydes and therefore limit cytotoxicity (*Pony et al., 2020*). However, direct evidence for the functional role of this putative channel is still lacking. In order to more fully understand the extended spirosome of *C. thermocellum*, we employed multiple computational approaches to analyze its structure. First, we used MOLE 2.0 to identify potential channels within the structure of the spirosome, specifically looking for channels that span between the ALDH and ADH active sites (*Pravda et al., 2018*). In all three structures (*E. coli* compact, *E. coli* extended, and *C. thermocellum* extended), as well as a SwissPDB model of the *C. thermocellum* compact structure, MOLE 2.0 identified a channel that could connect the two active sites. Compelling trends emerge when analyzing the composition of amino acids that line the channel compared to the amino acid composition of the whole protein for both *E. coli* and *C. thermocellum*. Overall, the channels are more highly conserved than the overall protein: the residues lining the extended channel have 92% homology, and the residues in the compact channel have 97% homology, whereas the overall protein homology is only 62%. When more AdhE sequences are included in the analysis, more variety is seen in the residues lining the channel (*Figure 6A*, *Figure 5—figure supplement 1*). Further examination of the individual amino acids also reveals important trends. First, there is only one lysine in the tunnel in all three structures, showing one of the largest $\log_2$-fold changes as compared to the composition of the full protein (*Figure 6B*). This may indicate that the channel is designed to contain reactive aldehydes while avoiding cross-linking with the amine group of the lysine. Second, there is a higher concentration of histidines in the channel in all the structures as compared to the full protein. This is potentially due to the tunnel passing by the catalytic iron in the ALDH domain, which is coordinated by three histidines. Notably, there is a difference in the number of phenylalanines in the channel when compared between the compact and extended structures. The extended structures contain a much higher density of phenylalanines as indicated by the $\log_2$-fold change. Finally, we focused on the cysteines present in these channels. In the compact state, only the catalytic cysteine from the ALDH domain is present in the channel. However, in both extended structures, there are two cysteines in the channel: the catalytic cysteine from the ALDH domain and the cysteine in the catalytic pocket from the ADH domain. This could point to the extended spirosome as being the relevant catalytic form because there is a potential path that efficiently connects the active sites.

As a complement to the insights gained from the static structures, MD simulations with bound acetaldehyde were conducted to investigate the dynamics of the aldehyde intermediate. We examined the effect of spirosome configuration (extended and compact) and species (*C. thermocellum* and *E. coli*) on residence time of acetaldehyde in the enzyme before escape to the solvent. The starting enzyme configurations for these simulations were the cryo-EM structure of the extended *C. thermocellum* spirosome, a homology model of the *C. thermocellum* compact spirosome (PDBID 6TQM serves as the template), and *E. coli* spirosomes in extended (6TQH) and compact (6TQM) forms (*Pony et al., 2020*). As in 6TQH and 6TQM, all simulation systems comprise two full AdhE units and two additional ADH domains on either side. The aldehyde intermediate is formed at the ALDH active site; thus, this constitutes the starting location for acetaldehyde in the simulations. Additional starting points midway between the ALDH and ADH catalytic sites were also generated based on the results from MOLE, namely midway along the MOLE tunnel where the channel was the widest (*Figure 7A and C*). More details for the system construction, acetaldehyde starting points, and simulation details are found in the Materials and methods. Three 200 ns MD simulations were conducted from each starting configuration. The primary conclusion from these is that the residence time for acetaldehyde when the spirosome is in the extended conformation is significantly longer than the compact conformation (*Figure 7E*, *Figure 6—figure supplement 1*). The significant difference between extended and compact forms presented in *Figure 7E* could be even larger given that many of the 200 ns simulations ended with acetaldehyde still present within the extended AdhE spirosomes (and thus the true residence time is longer than 200 ns). In addition, the trajectory of the aldehyde tracks quite closely with the channel found by MOLE (*Figure 7A and B*). Though acetaldehyde is sometimes observed to escape AdhE when started in the extended conformation, it typically samples a significant stretch of the path between the ALDH and ADH active sites, occasionally traversing the full

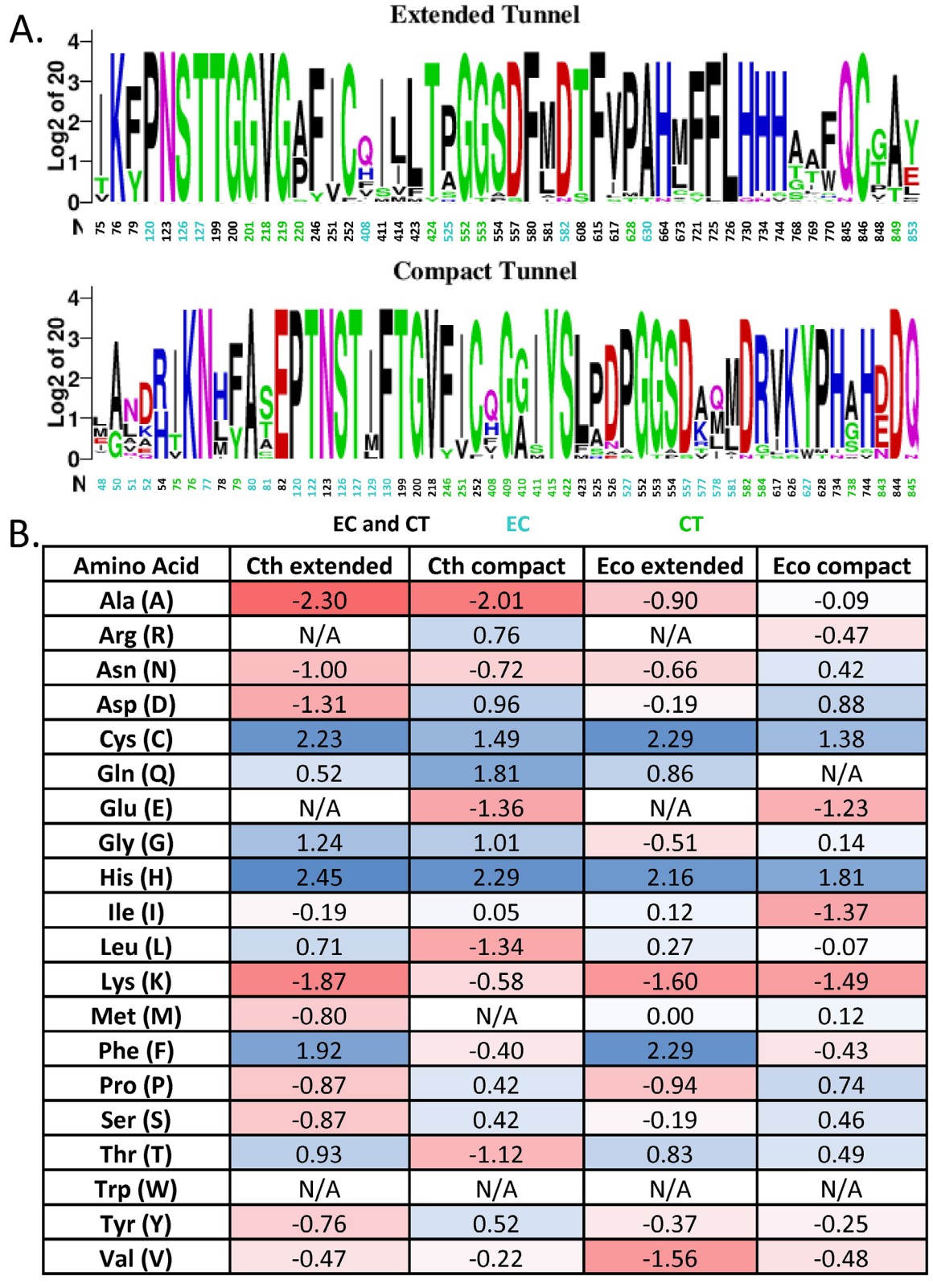

**Figure 6.** Analysis of the residues lining the spirosome channel. (**A**) Sequence homology of the channel from both *C. thermocellum* (Ct) and *E. coli* (Ec), residues only identified in the Ct channel are green and in the Ec only channel are cyan. (**B**) Log₂-fold difference between the residues that line the channel and the full-length protein. The scale of change is color-coded from deep red for fewer residues represented in the channel to deep blue for more residues represented in the channel.

*Figure 6 continued on next page*

**Figure supplement 1.** Molecular dynamics (MD) of aldehyde channeling in *E. coli*.

distance between the two active sites (*Figure 7 – Video 1*). In contrast, simulations initiated with the spirosome in the compact conformation typically see acetaldehyde leave within 4 ns (ALDH active site starting position) to 30 ns (starting point midway in the channel). For example, in the trajectory represented in *Figure 7D*, acetaldehyde leaves within 4 ns. We conducted analogous simulations with the *E. coli* AdhE and found the basic conclusions to be analogous to the *C. thermocellum* AdhE (*Figure 7*, *Figure 6—figure supplement 1*).

Taken together, these simulations effectively present a novel and dynamic understanding of intermediate aldehyde channeling in AdhE spirosomes and lend evidence to the hypothesis that the extended spirosome may be the dominant active complex for AdhE spirosomes.

Finally, we turned back to the variability of spirosome conformation observed by negative-stain TEM and searched more closely for the same variability in the cryo-EM data. Initial visual analysis was unable to discern between 2D projections of extended or compact conformations, as the difference was expected to be minute. The abundance of extended spirosomes relative to compact spirosomes confounded computational cross-correlation sorting while simultaneously making it difficult to visually identify compact projections. To address this, we sorted all particles selected by the crYOLO autopicker (which exhibits less bias than template or manual picking) using cryoDRGN (*Zhong et al., 2021*).

Correlation between particles visualized graphically in the Uniform Manifold Approximation and Projection (UMAP) (*Figure 8A*) indicated the existence of at least four distinct spirosome conformations. Volumes generated from these particle subgroups indicated the presence of extended, compact, and at least two major intermediate conformations (*Figure 8B*). We attempted to refine the isolated classes from cryoDRGN but encountered extended spirosome contamination in each of the classes. However, we continued searching for compact spirosome projections by generating 2D templates from the

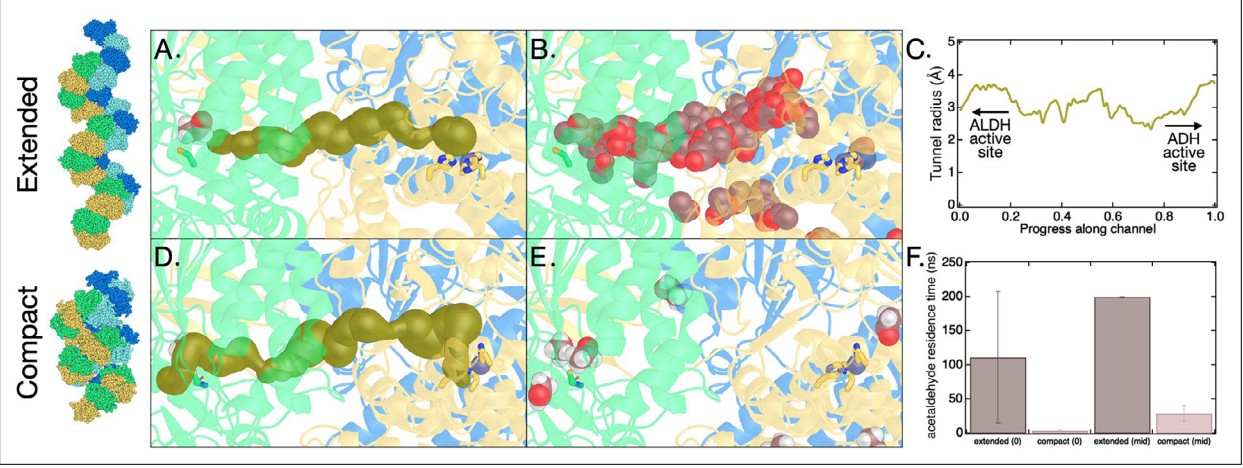

**Figure 7**. Molecular dynamics (MD) of aldehyde channeling in *C. thermocellum* aldehyde-alcohol dehydrogenase (AdhE) spirosome. (**A**) Starting configuration for MD simulation of the *C. thermocellum* extended spirosome structure overlaid with the channel connecting the aldehyde dehydrogenase (ALDH) and alcohol dehydrogenase (ADH) active sites as determined by MOLE (shown in gold spheres) (*Pravda et al., 2018*). Also shown are C252 (representing the ALDH active site, green sticks), three histidine residues that coordinate the divalent metal at the ADH active site (His664, His730, His744, yellow sticks), zinc ion (purple sphere), the starting location for acetaldehyde (violet spheres), and the tertiary AdhE structure (each AdhE molecule colored distinctly). (**B**) Same as panel A, except the MOLE tunnel is removed, and acetaldehyde location from MD simulation is shown every 1 ns for 160 ns. In this simulation, the acetaldehyde molecule exits the enzyme into solution after about 125 ns. (**C**) Representations are the same as in A, but here the channel shown is determined by MOLE for the compact spirosome structure (*E. coli* 6TQM and aligned with *C. thermocellum* AdhE homology structure). (**D**) Same as panel C, except MOLE tunnel is replaced with acetaldehyde location from MD simulation, shown every 1 ns. In this simulation, the acetaldehyde molecule exits the enzyme into solution at about 4 ns. (**E**) Average residence time for acetaldehyde in the channel before exiting AdhE in MD simulations for two different starting configurations for the AdhE spirosome (extended and compact) and acetaldehyde (at ALDH active site and midway along the channel). Error bars are the standard deviation for simulations performed in triplicate.

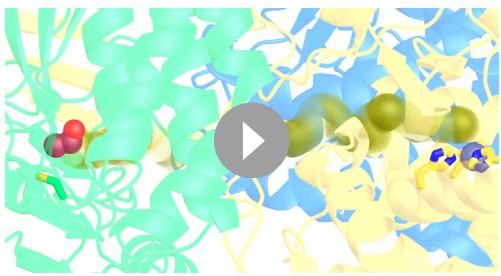

**Video 1.** Molecular dynamics (MD) of aldehyde channeling. Example of MD simulation of *C. thermocellum* extended spirosome with the channel connecting the aldehyde dehydrogenase (ALDH) and alcohol dehydrogenase (ADH) active sites as determined by MOLE (*Pravda et al., 2018*) shown in gold spheres. Also shown are C252 (representing the ALDH active site, green sticks), three histidine residues that coordinate the divalent metal at the ADH active site (His664, His730, His744, yellow sticks), zinc ion at ADH active site (purple sphere), and the tertiary AdhE structure (each AdhE molecular colored distinctly). Acetaldehyde (violet spheres) is initiated at the ALDH active site and throughout the simulation for 130 ns. In this simulation, the acetaldehyde molecule exits the enzyme into solution after about 125 ns. The movie was created in PyMOL (The PyMOL Molecular Graphics System, version 2.0 Schrödinger, LLC) and the positions of each residue are smooth using the 'smooth' command with two passes and a window size of two. This same starting configuration was run in triplicate, with retention times of 9 and 200 ns in the other two simulations.

https://elifesciences.org/articles/96966/figures#video1

*E. coli*, compact AdhE spirosome model (PDBID 6AHC) using CryoSPARC (*Figure 8C*; *Kim et al., 2019*; *Punjani et al., 2017*). These templates were then used as a visual reference, which did indeed reveal the presence of compact 2D projections within the *C. thermocellum* cryo-EM data (6.7% of picked particles were compact) that were previously overlooked (*Figure 8D*). The subsequent reconstruction using these particles yielded a 3.93 Å density of the compact AdhE spirosome, although there was evidence of streaking due to preferred orientation issues and low particle numbers. Although the streakiness prevented the successful fitting of a high-resolution molecular model, the density map corroborates the domain orientation of a compact spirosome, as modeled by SWISS-MODEL. There are some intriguing insights into differences between our compact ultrastructure compared to previously reported compact AdhE spirosomes isolated from different species (*Kim et al., 2019*; *Pony et al., 2020*; *Cho et al., 2021*). Most notably, the *C. thermocellum* AdhE does not achieve as compact a state, even when isolating the most compact particles. Moreover, the reconstruction of a compact spirosome density corroborates that compact spirosomes observed via negative staining are also present in near-native conditions (cryogenically preserved samples). This data confirms that spirosomes have the potential to exist in a spectrum of states that can be captured in a single cryo-EM grid without any additional cofactors to induce conformational changes.

## Conclusions

In this study, we presented and analyzed a high-resolution structure of the AdhE spirosome from *C. thermocellum*. In comparison to the *apo E. coli* spirosome structure, which purifies in the compact conformation, the *apo C. thermocellum* spirosomes are primarily extended (*Kim et al., 2019*; *Pony et al., 2020*). Our structural comparison identifies the extended conformation as the more structurally stable form in both the case of *E. coli* and *C. thermocellum*, which disagrees with the reality of the compact spirosome being the stable, *apo* conformation from *E. coli*. The evidence for the extended spirosome as the more stable form includes the higher number of salt bridges and hydrogen bonds between molecules in the extended as compared to the compact. Furthermore, extended spirosomes have a larger buried surface area in the dimer interface than the compact spirosomes (5000 Å$^2$ compared to 3800 Å$^2$, respectively), imparting additional stability. Although we could not structurally identify exact features that would predict which conformation was preferred, we have evidence from negative-stain EM experiments that conformation does not rely solely upon the environment: *C. thermocellum* spirosomes are extended in both endogenous and exogenous expression conditions. Therefore, there must be some intrinsic structural elements that contribute to the *apo* state of the spirosome, although we have not found that element in this study. This could potentially be due to the differences between Gram-positive and Gram-negative bacteria. In previous studies, compact spirosomes have only been isolated from Gram-negatives while solely extended spirosomes have been isolated from Gram-positives. Furthermore, while the compact spirosomes can transition to extended in the presence of cofactors, the reverse has not been previously observed with an extended spirosome (*Matayoshi and Oda, 1985*; *Kim et al., 2019*; *Pony et al., 2020*; *Laurenceau et al., 2015*).

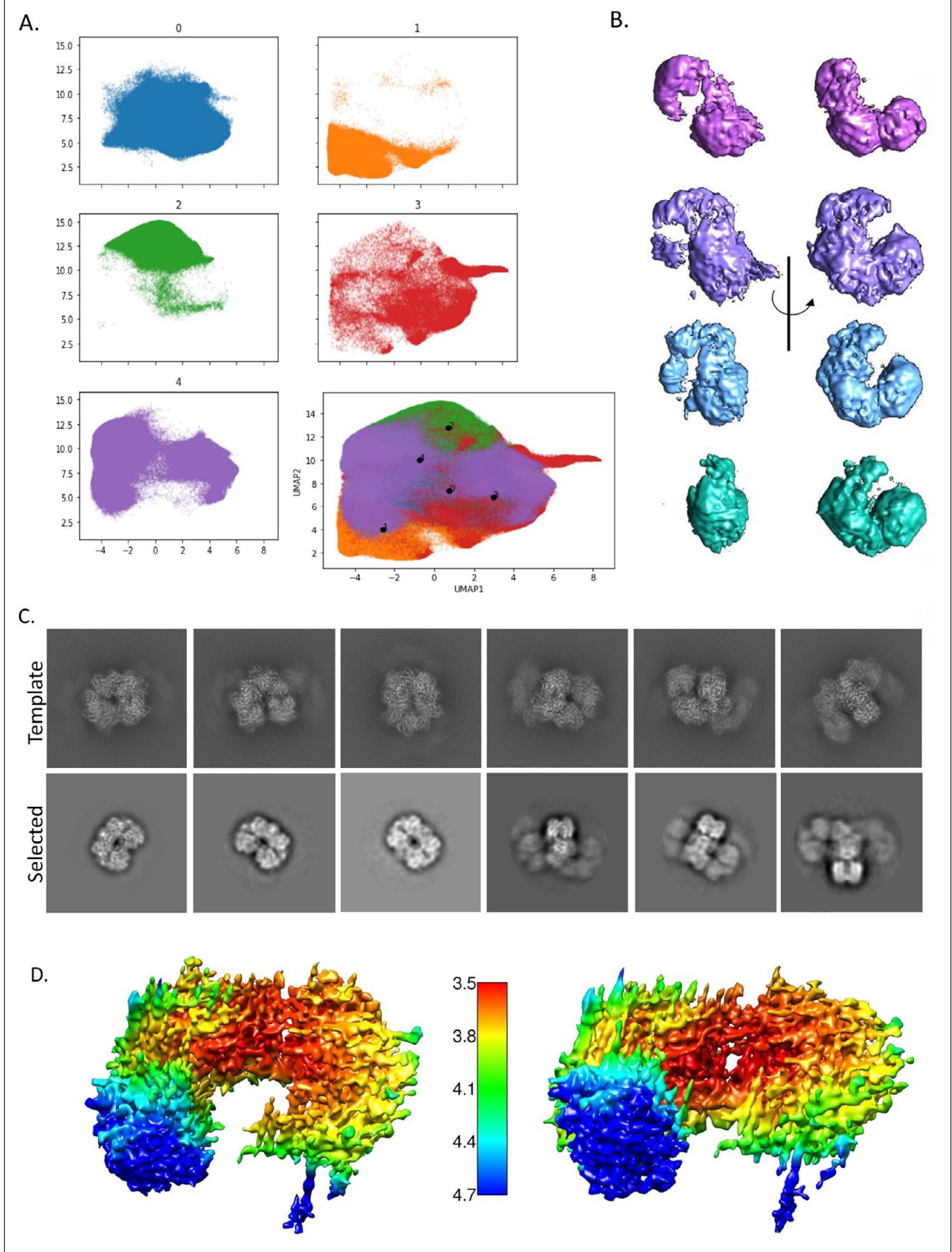

**Figure 8.** CryoDRGN results show that there is conformational heterogeneity in the sample. (**A**) Uniform Manifold Approximation and Projection (UMAP) representation of the particles shows a continuous heterogeneity (bottom right corner). Colored UMAP graph shows the division of the particles into five classes, broken into their individual map based on color. (**B**) One representative density for each group (excepting the junk class) shown at two angles to illustrate the movement of the spirosome (purple – extended, lavender – intermediate 1, light blue – intermediate 2, teal – compact). (**C**) Back-projected

*Figure 8 continued on next page*

*Figure 8 continued*

template classes generated in CryoSPARC from PDB ID 6AHC (top) and the matching classes selected from our data (bottom). (**D**) Two angles of the final compact spirosome density shown with local resolution coloring as indicated by the scale bar in the center.

Here, we identified compact spirosomes in our study, although they were largely outnumbered by the more-preferred extended spirosome, which further indicates that the extended structure is more stable for *C. thermocellum*, regardless of expression host. While these compact spirosomes could result from expression in *E. coli*, we also identified compact spirosomes in a native *C. thermocellum* lysate, which would not have similar contamination issues. Close examination of the structure revealed a potential novel allosteric NAD(P)H binding site that lies between two monomers of AdhE, identified by a characteristic NAD(P)H binding motif. The GxGxxG motif is highly conserved across 1000 AdhE sequences, indicating its importance in the protein. Further studies will be necessary to fully understand the role of this motif.

The second conclusion of this paper is that intermediate channeling is possible and may be the primary purpose of large, spirosome ultrastructures. An intriguing finding from prior research is that the spirosome is necessary for the acetyl-CoA to acetaldehyde reaction in the ALDH domain, yet unneeded for the reverse reaction or for ADH activity (*Kim et al., 2019*; *Pony et al., 2020*). Therefore, a channel might be necessary to funnel acetaldehyde away from the ALDH active site to prevent product inhibition in the ALDH site. Prior research identified a similar channel in *E. coli* AdhE spirosomes yet did not confirm its role nor characterize the movement of intermediates within the channel (*Pony et al., 2020*). Here, we identified an enclosed tunnel between an ALDH domain and an ADH domain of two separate proteins in *C. thermocellum* and used MD to examine possible aldehyde channeling. While the channel can be found in both the compact and extended structures, we see that the channel in the compact conformation has areas that are exposed to the cytosol, whereas the extended channel is entirely enclosed. This further supports the idea that the extended spirosome is the active form of AdhE (*Kim et al., 2020*). MD simulations further corroborate this, showing that the aldehyde intermediate has a much longer occupation time in the channel of the extended form. Future mutagenesis studies will be needed to confirm whether the spirosome exists to control the reaction flux in high-reactant conditions.

The *C. thermocellum* AdhE structure, in combination with MD and comparative analysis with the *E. coli* structures, establishes a baseline for further studies using mutagenesis to increase the efficacy of the protein. This work provides insight as to why the spirosome seems to be necessary for only one direction of the reaction, as well as opens new and interesting questions as to why certain variants of AdhE natively exist in different conformations. It also provides a template that could be replicated in other systems where toxic intermediates need to be sequestered to increase the production of valuable biochemicals.

## Materials and methods

### Gene cloning in *E. coli*

*E. coli* and *C. thermocellum* AdhE genes fused with N-terminal His-tag and TEV site used in this study were amplified from their genomic DNA and inserted into pET28b vector. Gibson Assembly Cloning Kit (NEB, Ipswich, MA, USA) and 5-alpha *E. coli* strain (NEB, Ipswich, MA, USA) were used for gene cloning. Gene cloning protocols reported previously were followed (*Xu et al., 2003*). Primers, plasmids, strains, and sequences of the expressed genes used in the study are listed in *Figure 1*, *Supplementary file 1A-B*.

### Gene expression in *E. coli*

The generated plasmids used for AdhE gene expression described above were transformed into *E. coli* BL21(DE3) strain (NEB, Ipswich, MA, USA). The transformants were grown aerobically in LB medium supplemented with kanamycin at 37°C and 225 rpm overnight (seed culture). The seed culture in a 1:50 ratio was inoculated into LB medium containing the same antibiotic, and the cells were grown under the same growth conditions until optical cell density $OD_{600}$ reached 0.8. IPTG (0.3 mM) was added to the culture, and cells were grown overnight at 16°C and 225 rpm after the culture was

cooled on ice for 3 hr. The cells were collected by centrifuging 8000×*g* for 10 min and used for further protein extraction and purification.

## Protein isolation and purification

Cell lysis was performed in HisA buffer (50 mM Tris pH 7.5, 400 mM NaCl, 20 mM imidazole, 1% glycerol), 20 mg of lysozyme, 1 µL of DNase I, and one protease inhibitor tablet, all set to rock at room temperature for 1 hr. Lysed cells were spun down at 15,000×*g*, and the clarified lysate was loaded onto a His-trap column for affinity purification. The tagged protein was eluted in HisB (50 mM Tris pH 7.5, 400 mM NaCl, 400 mM imidazole, 1% glycerol), then applied to an S200 size exclusion column for a final step of purification in 50 mM Tris pH 7.5 and 150 mM NaCl. Protein was stored at 4°C before either negative stain or cryo-EM grid preparation.

## Native *C. thermocellum* cell growth and isolation

Fermentations were carried out using *C. thermocellum* strain DSM1313. Cultures were grown in Medium for Thermophilic Clostridia (MTC), using stock solutions to combine with water and the desired carbon source to reach 1× concentration. The medium was prepared as described with cellobiose as the carbon source (*Holwerda et al., 2012*).

Glycerol stocks were revived in 10 mL serum bottles that were prepared in an anaerobic chamber containing 85% $N_2$, 10% $CO_2$, and 5% $H_2$. Seeds were grown to 0.8–1.0 $OD_{600}$ at 55°C and transferred to 100 mL serum bottles. These cultures were grown to an $OD_{600}$ of 0.8–1.0 in approximately 12 hr before the entire contents (10% vol/vol) were inoculated into 5 g/L cellobiose MTC medium in process control bioreactors for a final working volume of 1 L. All growth medium was prepared at pH 7.0 and maintained at this level for all bioreactor experiments using 2 N KOH. Temperature was maintained at 55°C with 50 rpm agitation for bottles and bioreactors. An $N_2$ environment was maintained through the bioreactor headspace at a rate of 30 mL/min.

Cultures were grown until maximum cell density was reached around 1.2 $OD_{600}$ in approximately 8 hr. The entire bioreactor contents were harvested into centrifuge bottles and centrifuged at 8000 rpm for 20 min. The supernatant was removed by decanting. The pellet was then washed with distilled water and centrifuged again under the same conditions. The wash fraction was removed by decanting, and the residual pellet was frozen until use in experiments.

Cells were lysed using a BeadBeater with a 30 s on/30 s off pattern for 5 min in SEC buffer (50 mM HEPES pH 7.4, 150 mM NaCl) with 1 µL of DNase I and one tablet of protease inhibitor. Lysate was spun down at 15,000×*g* to pellet any extra cell debris, then concentrated and passed through a 0.22 µm filter before being loaded directly onto a Superose 6 size exclusion column. The fractions that had a signal on the FPLC trace were stored at 4°C before negative stain grid preparation.

## Negative-stain TEM sample preparation and data collection

All AdhE proteins were screened for optimal concentrations for negative staining – finding that 0.05 mg/mL was the ideal concentration for protein separation. Grids were glow discharged for 20 s at 10 mA before the sample was applied to the grid to incubate for 1 min. The excess sample was blotted away before the grid was dipped in a droplet of 2% aqueous uranyl acetate (UA) for 15 s. Excess UA was blotted away, and grids were left to dry before storage at room temperature before TEM screening. Grids of the *apo* nCthAdhE, CthAdhE, and EcoAdhE were viewed on a Tecnai T20 (FEI/Thermo Fisher Scientific) TEM at ×50,000 magnification, with 6×6 montages collected using SerialEM (*Mastronarde, 2005*). Grids of the CthAdhE and EcoAdhE with added reactants were viewed using a Talos L120C TEM at ×36,000 magnification, using an automated collection strategy through Leginon (*Suloway et al., 2005*).

## Negative staining data analysis

Spirosome length and overall conformation were manually analyzed in ImageJ (FIJI) (*Schindelin et al., 2012*). Image pixel size was used to calibrate the scale bar before the line tool was used to trace and measure the length of the spirosome. Compact vs extended was determined by eye for at least 100 and up to 1200 spirosomes for statistical relevance. Statistical relevance of differences in the length of spirosomes was calculated using a Mann-Whitney U test; all were found to be statistically different from each other.

## Cryo-EM sample preparation and data collection

C-Flat 1.2/1.3 400 mesh grids were glow discharged for 10 s before 0.4–0.5 mg/mL of CthAdhE was applied. Grids were blotted for 3 s using a CP3 Cryoplunge 3 (Gatan Ametek Inc) before being plunged into liquid ethane held at –168°C. Preliminary data was collected on a 200 kV Talos Arctica (Thermo Fisher Scientific) cryo-TEM with K3 direct electron detector (Gatan Ametek Inc) using a total electron dose of 90 e$^-$/Å$^2$ and an exposure time of 3 s. The dataset used in the final extended *C. thermocellum* spirosome reconstruction was collected on a 300 kV Titan Krios cryo-TEM with a Falcon 4 direct electron detector (Thermo Fisher Scientific) using a total electron dose of 60 e$^-$/Å$^2$ and an exposure time of 6.49 s. Finally, the data used for the compact CthAdhE analysis was collected on a 300 kV Titan Krios cryo-TEM with a Falcon 4 (Thermo Fisher Scientific) direct electron detector using a total electron dose of 60 e$^-$/Å$^2$ and an exposure time of 4.42 s.

## Cryo-EM data analysis

Structural analysis was done in Relion 3.1, originally starting with helical processing to create a 3D template for autopicking in single particle analysis, resulting in a 3.8 Å structure that was used for initial model building (*Figure 2*, *Supplementary file 1C*, *Figure 2—figure supplements 2 and 3*; *Nakane et al., 2020*). Data from the Krios was also processed in Relion 3.1 using single particle analysis (*Nakane et al., 2020*). Using a soft mask around the edge of the protein, a final map was generated that had an FSC 0.143 of 3.28 Å (*Figure 2*, *Supplementary file 1C*). The CCP-EM software suite was used to dock a homology model of the *C. thermocellum* structure into the density (*Burnley et al., 2017*). Coot was used to refine the structure iteratively with CCP-EM refinement and validation (*Emsley and Cowtan, 2004*). The best structure was taken to Phenix, where it was docked into the auto-sharpened map generated by Phenix (*Liebschner et al., 2019*). Final rounds of refinement and validation resulted in the structure deposited to the PDB as 8UHW, and the final map was deposited to the EMDB as EMD-44284.

## Sequence alignments and analysis

25 sequences previously identified in literature were aligned in Clustal Omega (*Kessler et al., 1992*; *Matayoshi and Oda, 1985*; *Extance et al., 2013*; *Extance et al., 2016*; *Laurenceau et al., 2015*; *Sievers et al., 2011*). The percentage of conservation was calculated by hand, and the alignment logos were generated with WebLogo (*Crooks et al., 2004*). For the 1000-sequence dataset, sequences were identified by homology on the JGI IMG database (*Chen et al., 2023*). All relevant sequences were input into the MAFFT online service (*Katoh et al., 2019*; *Kuraku et al., 2013*). Protein length settings were used to exclude sequences that only aligned with one domain before final analysis of the consensus sequences.

## MD simulations

Classical MD simulations of 200 ns in length were run in triplicate for the following enzymes: (1) *C. thermocellum* AdhE in extended spirosome form, (2) *C. thermocellum* AdhE in compact form, (3) *E. coli* AdhE in extended form, and (4) *E. coli* AdhE in compact form. With the exception of *C. thermocellum* in compact form, the starting structures for the enzymes come from cryo-EM structures. The *E. coli* extended and compact forms originate with the structures from Pony et al. (PDB codes 6TQH and 6TQM, respectively; *Pony et al., 2020*). The *C. thermocellum* compact spirosome was constructed as a homology model by SWISS-MODEL with 6TQM as the template (*Pony et al., 2020*; *Waterhouse et al., 2018*). The simulation system comprises two full AdhE units and two additional ADH domains on either side, as in 6TQH and 6TQM. Where they appear in the cryo-EM structures, NAD+ and NADH molecules are removed. All three cryo-EM structures contain divalent iron bound at ADH active sites. The position of these cations is maintained, but iron is swapped for zinc in each case. We have chosen to include zinc in each ADH active site, rather than iron, for two primary reasons. The first is that parameters for iron are not included in the standard CHARMM force field. Second, it has been shown that ADH activity of ADHE can be stimulated in cell extracts to varying degrees, including zinc (*Extance et al., 2013*). In fact, divalent zinc has been shown to have higher ADH activity than divalent iron in at least one study, leading the authors to conclude that this class of ADH structures should be described as 'metal ion-dependent' rather than strictly iron- or zinc-specific (*Extance et al., 2013*).

In the biofuels context, i.e., producing alcohols from acyl-CoA molecules, the aldehyde intermediate is formed at the ALDH active site from acyl-CoA and then travels to the ADH active site where it is reduced to an alcohol. Thus, for our MD simulations, positioning the acetaldehyde intermediate at the ALDH active site most closely approximates the physical situation for intermediate channeling. The placement of acetaldehyde at the ALDH active site was determined by alignment with 5JFM (*Zarzycki et al., 2017*). Additional starting points midway between the ALDH and ADH catalytic sites were generated based on the results from MOLE, namely choosing starting coordinates midway along the MOLE tunnel where the channel was widest (*Figure 7*, *Figure 6—figure supplement 1*; *Pravda et al., 2018*).

In general, protonation states were determined by the output from the H++ server (http://biophysics.cs.vt.edu/H++) at pH 6.0, consistent with the assays performed by Extance et al. In addition, the catalytic cysteine residues in ALDH domains (C252 in *C. thermocellum and* C246 in *E. coli*) are deprotonated. The overall charge on the enzyme domains (two full AdhE molecules and two additional ADH domains) with these protonation states is –42 for *C. thermocellum* systems and –28 for *E. coli* systems; a corresponding number of sodium ions were added to the solution phase to neutralize the overall system. Neither *C. thermocellum* nor *E. coli* AdhE contains any disulfide bonds.

Furthermore, to investigate the dynamics of water molecules at the ADH active site, 80 ns MD simulations were performed in duplicate of the *C. thermocellum* extended structure with NADH bound at the ADH active site. Zinc molecules are positioned at each ADH active site, as above.

All simulations were built using CHARMM version 44a1 and simulated with the CHARMM36 force field for the protein and TIP3P potential for water molecules (*Best et al., 2012*; *Brooks et al., 2009*; *Jorgensen et al., 1983*). Topologies and force field parameters for acetaldehyde and NADH come from the CHARMM generalized force field (CGenFF) version 4.1 (*Vanommeslaeghe et al., 2010*; *Yu et al., 2012*).

## CryoDRGN analysis

To evaluate structural heterogeneity evident in the cryo-EM data, 959,561 potential spirosome particles were sorted into 2D classes using RELION 3.1 and were exported for analysis with cryoDRGN 0.3.4 (*Zhong et al., 2021*; *Nakane et al., 2020*). Particles were downsampled to a 64 px box size (5.28 Å/px) and analyzed for 50 epochs with the standard latent space parameter of 8, at which point training converged. Inspection of 20 volumes generated by k-means clustering and resulting UMAP visualization of latent space of particle projections suggested the presence of four major conformational classes, which were segmented by Gaussian mixture modeling (GMM, k=4). Of them, only major classes indicative of spirosome formation were retained for further analysis. A fifth class ('Other') of unidentifiable 'junk' particles and dimers was also identified, and those particles were discarded from further processing. The remaining 858,511 spirosome particles were scaled to 128 px box size (2.64 Å/px) and again analyzed for 50 epochs. Major classes indicated by the UMAP clusters were attributed to extended, compact, and transitional state spirosomes by visual confirmation of 3D density volumes generated by k-means clustering and filtered by GMM (k=5).

## CryoSPARC template generation and use

The cryoSPARC 'Create Templates' utility was used to back-project 50 2D classes of the compact *E. coli* AdhE spirosome (*Pony et al., 2020*; *Punjani et al., 2017*). For posterity, 50 back-projected 2D classes of the extended *E. coli* AdhE spirosome were also generated to serve as a comparison (*Pony et al., 2020*). Visual inspection indicated sufficient difference between 2D projections of both conformations. The 2D templates of the compact structure were used as a visual reference by which compact classes were hand-selected from the heterogeneous compact model building described above. This resulted in the selection of 64,977 particles that were determined to be in the compact state, which were processed in Relion 3.1 as described above. Because the heterogeneous dataset was lacking in some projections predicted by the template generation, the resulting 3D density of the compact *C. thermocellum* AdhE suffered from some preferred orientation and ultimately achieved a 3.93 Å resolution, as determined by the Relion gold standard FSC.

## Acknowledgements

Portions of this work were performed at the CU Anschutz Medical Campus Cryo-EM Facility and the SLAC S2C2 Cryo-EM Facility. We thank both facilities for the use of their microscopes and related equipment. Computer time was provided by the National Renewable Energy Laboratory Computational Sciences Center supported by the DOE Office of EERE under contract number DE-AC36-08GO28308. This material is based upon work supported by the Center for Bioenergy Innovation (CBI), U.S. Department of Energy, Office of Science, Biological and Environmental Research Program under Award Number ERKP886.

## Additional information

### Funding

| Funder | Grant reference number | Author |
| --- | --- | --- |
| U.S. Department of Energy | DE-AC36-08GO28308 | Samantha J Ziegler<br>Brandon Knott<br>Josephine N Gruber<br>Neal N Hengge<br>Qi Xu<br>Daniel G Olson<br>Yannick Bomble |
| Oak Ridge National Laboratory | ERKP886 | Samantha J Ziegler<br>Brandon C Knott<br>Josephine N Gruber<br>Neal N Hengge<br>Qi Xu<br>Daniel G Olson<br>Eduardo E Romero |

The funders had no role in study design, data collection and interpretation, or the decision to submit the work for publication.

### Author contributions

Samantha J Ziegler, Brandon C Knott, Conceptualization, Investigation, Visualization, Methodology, Writing – original draft, Writing – review and editing; Josephine N Gruber, Investigation, Visualization, Writing – original draft, Writing – review and editing; Neal N Hengge, Qi Xu, Eduardo E Romero, Lydia-Marie Joubert, Methodology, Writing – review and editing; Daniel G Olson, Conceptualization, Writing – review and editing; Yannick J Bomble, Conceptualization, Supervision, Writing – original draft, Writing – review and editing

### Author ORCIDs

Samantha J Ziegler ⓘ https://orcid.org/0000-0003-2480-513X
Brandon C Knott ⓘ https://orcid.org/0000-0003-3414-3897
Eduardo E Romero ⓘ https://orcid.org/0000-0002-0793-9948
Yannick J Bomble ⓘ https://orcid.org/0000-0001-7624-8000

Reviewer #1 (Public Review): https://doi.org/10.7554/eLife.96966.3.sa1
Reviewer #2 (Public Review): https://doi.org/10.7554/eLife.96966.3.sa2
Author response https://doi.org/10.7554/eLife.96966.3.sa3

## Additional files

### Supplementary files

Supplementary file 1. Tables of data associated with this study. (A). Sequences of AdhE primers. Nucleotide sequences of the primers used in this study. (B). Sequences of the AdhE constructs. Amino acid sequences of the two proteins, *E. coli* and *C. thermocellum* AdhE, used in this study. (C). Data collection and processing. The statistics describing the cryo-electron microscopy (cryo-EM)

data processing performed in this study.

MDAR checklist

## Data availability

PDB ID 8UHWEMDB EMD-42284.

The following datasets were generated:

| Author(s) | Year | Dataset title | Dataset URL | Database and Identifier |
|---|---|---|---|---|
| Ziegler SJ, Gruber JN | 2024 | The structure of the Clostridium thermocellum AdhE spirosome | https://doi.org/10.2210/pdb8uhw/pdb | Worldwide Protein Data Bank, 10.2210/pdb8uhw/pdb |
| Ziegler SJ, Gruber JN | 2024 | The structure of the Clostridium thermocellum AdhE spirosome | https://www.ebi.ac.uk/emdb/EMD-42284 | Electron Microscopy Data Bank, EMD-42284 |

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
