## [Editor Report · eLife assessment]

This work presents **valuable** information on the structure of the spirosome's native extended conformation as the active form of the aldehyde-alcohol dehydrogenase (AdhE) enzyme. The evidence is **solid**, although the work does not provide a mechanistic understanding of the function and dynamics of AdhE.

---

## [Referee Report · Reviewer #1 (Public Review)]

Clostridium thermocellum serves as a model for consolidated bioprocessing (CBP) in lignocellulosic ethanol production. The primary ethanol production pathway involves the enzyme aldehyde-alcohol dehydrogenase (AdhE), which exhibits complex regulation, forming long oligomeric structures known as spirosomes.

The present study describes the cryo-EM structure of C. thermocellum AdhE, resolved at 3.28 Å resolution. By integrating cryo-EM data with molecular dynamics simulations, this study showed that the aldehyde intermediate resides longer in the channel of the extended form, supporting the mechanistic model in which the extended spirosome conformation represents the active form of AdhE.

These findings advance the understanding of the function and regulation of AdhE, a key enzyme involved in the ethanol biosynthesis pathway in Clostridium thermocellum, a model organism for ethanol production in consolidated bioprocessing.

---

## [Referee Report · Reviewer #2 (Public Review)]

Summary:

The manuscript by Ziegler et al, entitled “Structural characterization and dynamics of AdhE ultrastructure from *C. thermocellum*: A containment strategy for toxic intermediates?” presents the atomic resolution cryo-EM structure of *C. thermocellum* AdhE showing that it show dominantly an extended form while *E. coli* AdhE shows dominantly a compact form. With comparative analysis of their *C. thermocellum* structure and the previous *E. coli* AdhE structure, they tried to reveal the mechanism by which *C.thermocellum* and *E. coli* show different dominant conformations. In addition, they also analyzed the substrate channel by comparative and computational approaches. Lastly, their computational analysis using CryoDRGN reveals conformational heterogeneity in the sample. Despite this the manuscript is very descriptive and does not provide a mechanistic understanding by which AdhE works, this work will provide structural frame works to further investigate the function and mechanism of AdhE dynamics.

Strengths:

This manuscript provides the first *C. thermocellum* (Ct) AdhE structure and comparatively analyzed this structure with *E. coli* AdhE.

Weaknesses:

This work is very descriptive and does not provide mechanistic understanding of the function and dynamics of AdhE.

---

## [Author Response]

The following is the authors’ response to the original reviews.

**Public Reviews:**

**Reviewer #1 (Public Review):**
Summary:*Clostridium thermocellum* serves as a model for consolidated bioprocess (CBP) in lignocellulosic ethanol production, but yet faces limitations in solid contents and ethanol titers achieved by engineered strains thus far. The primary ethanol production pathway involves the enzyme aldehydealcohol dehydrogenase (AdhE), which forms long oligomeric structures known as spirosomes, previously characterized via the 3.5 Å resolution *E. coli* AdhE structure using single-particle cryoEM. The present study describes the cryo-EM structure of the *C. thermocellum* ortholog, sharing 62% sequence identity with E. coli AdhE, resolved at 3.28 Å resolution. Detailed comparative structural analysis, including the Vibrio cholerae AdhE structure, was conducted. Integrating cryoEM data with molecular dynamics simulations indicated that the aldehyde intermediate resides longer in the channel of the extended form, supporting the hypothesis that the extended spirosome represents the active form of AdhE.Strengths:The study conducts a comprehensive structural comparative analysis of oligomerization interfaces and the acetaldehyde channel across compact and extended conformations. Structural and computational results suggest the extended spirosome as the most likely active state of AdhE.Weaknesses:The overall resolution of the *C. thermocellum* structure is similar to the *E. coli* ortholog, which shares 62% sequence identity, and the oligomerization interfaces and the acetaldehyde channel were previously described.
**Reviewer #2 (Public Review):**
Summary:The manuscript by Ziegler et al, entitled 'Structural characterization and dynamics of AdhE ultrastructure from *Clostridium thermocellum*: A containment strategy for toxic intermediates?" presents the atomic resolution cryo-EM structure of *C. thermocellum* AdhE showing that it show dominantly an extended form while *E. coli* AdhE shows dominantly a compact form. With comparative analysis of their *C. thermocellum* structure and the previous *E. coli* AdhE structure, they tried to reveal the mechanism by which *C. thermocellum* and *E. coli* show diXerent dominant conformations. In addition, they also analyzed the substrate channel by comparative and computational approaches. Lastly, their computational analysis using CryoDRGN reveals conformational heterogeneity in the sample. Although this manuscript suggests a potential mechanism of the diXerent features of AdhEs, this manuscript is very descriptive and does not provide suXicient data to support the authors' conclusions, which may be due to the lack of experimental data to support their findings from the computational analysis.Strengths:This manuscript provides the first *C. thermocellum* (Ct) AdhE structure and comparatively analyzed this structure with *E. coli* AdhE.Weaknesses:Their main conclusions obtained mostly by computational and comparative analysis are not supported by experimental data.
**Reviewer #3 (Public Review):**
This study describes the first structure of Gram-positive bacterial AdhE spirosomes that are in a native extended conformation. All the previous structures of AdhE spirosomes obtained come from Gram-negative bacterial species with native compact spirosomes (*E. coli*, *V. cholerae*). In *E. coli*, AdhE spirosomes can be found in two diXerent conformational states, compact and extended, depending on the substrates and cofactors they are bound to.The high-resolution cryoEM structure of the extended *C. thermocellum* AdhE spirosomes produced in *E. coli* in an apo state (without any substrate or cofactors) is compared to the *E. coli* extended and compact AdhE spirosomes structures previously published. The authors have modeled (in Swiss-Model) the structure of compact *C. thermocellum* AdhE spirosomes, using *E. coli* compact AdhE spirosome conformation as a template, and performed molecular dynamics simulations. They have identified a channel in which the toxic reaction intermediate aldehyde could transit from the aldehyde dehydrogenase active site to the alcohol dehydrogenase active site, in an analogous manner to *E. coli* spirosomes. These findings are in line with the hypothesis that the extended spirosomes could correspond to the active form of the enzyme.In this work, the authors speculate that the *C. thermocellum* AdhE spirosomes could switch from the native extended conformation to a compact conformation, in a way that is inverse of *E. coli* spirosomes. Although attractive, this hypothesis is not supported by the literature. Amazingly, in some Gram-positive bacterial species (*S. pneumoniae, S. sanguinis or C. di8icile*...), AdhE spirosomes are natively extended and have never been observed in a compact conformation. On the opposite, *E. coli* (and other Gram-negative bacteria) native AdhE spirosomes are compact and are able to switch to an extended conformation in the presence of the cofactors (NAD+, coA, and iron). The data presented as they are now are not convincing to confirm the existence of *C. thermocellum* AdhE spirosomes in a compact conformation.
**Recommendations for the authors:**

**Reviewer #1 (Recommendations For The Authors):**
Major points:(1) The claim of achieving the highest resolution AdhE structure lacks strong support since the *E. coli* structure was solved at 3.5A, whereas the *C. thermocellum* was solved at 3.28A. Conducting a local resolution analysis could provide insights into distinct structural interpretations, enhancing the strength of the claim.

We have modified the sentence claiming this as the highest resolution AdhE structure to say, “In this study, we presented and analyzed a high-resolution structure of the AdhE spirosome from *C. thermocellum*.” We have included the local resolution map in Figure 2C – all structural analysis was performed in regions from the center of the molecule, where the highest resolution information was determined.

(2) The comparative structural analysis of the oligomerization interface is thorough, yet it could benefit from greater conciseness. Focusing on highlighting major findings would streamline the presentation and enhance clarity.

We altered a few places in the comparative structural analysis in response to other reviewers. We also divided the main structure section into two subsections (spirosome interfaces and AdhE active sites) to enhance clarity.

**Reviewer #2 (Recommendations For The Authors):**
(1) The authors should change the tile containing "?". Does it mean that the conclusions that the authors made are still in question?

We have removed the question mark to indicate that our results point to a channeling mechanism.

(2) Figure 1B: Clarify Ct Fwd. Is this adding NADH, and Ct Rev adding NAD+?

This information is described in the text in lines 98-100. It is also at the bottom of figure 1B.

(3) Line 131: Please revise accordingly for clarity: "The extended dimer interface and The extended *E. coli* dimer interface".

This has been edited for clarity. We have added the following sentence resulting to indicate which interfaces that are being discussed: “Both the *E. coli* and *C. thermocellum* extended dimer interfaces bury ~5000 Å2. While the compact *C. thermocellum* compact dimer interface buries a similar surface area of ~4800 Å2, the *E. coli* dimer interface buries ~3800 Å2.”

(4) Line 133-136: Why that does not seem to be the case? These sentences are not clear what the authors exactly mean.

We altered the text to say, “One would expect the compact structure in *E. coli* to have a larger buried surface area due to it being the predominant form when it is examined without additives, but that is not the case; further corroborating that factors other than buried surface area must impact the *apo* state of the spirosome.” We hope this clarifies our intent.

(5) Line 138-145: The authors should provide a logic for how the diXerent distribution of the charged residues would change the form of AdhE. It may just be a diXerent distribution nothing to do with the conformational change.

After further analysis of the interface amino acid distribution, we agree that the distribution may have nothing to do with the conformational change. We have changed this section to end with the sentence “Analysis of the residues buried in these interfaces reveals that while many of the residues are identical in the *C. thermocellum* and *E. coli* extended structures, there are some diXerences in amino acid type distribution, although nothing that directly indicates control of conformer state (Supplemental Figure 3).”

(6) Line 169: Kim et al. è Cho et al.

We have corrected this error.

(7) Line 122-235: The whole section is just describing the diXerence between Ct and Ec AdhE suggesting that this diXerence may contribute to the conformational diXerence without any evidence. The author cannot say that the diXerences in the interface, active sites cofactor pockets, etc explain why two AdhE (Ct, Ec) have diXerent domain conformers unless they provide experimental data.

We did not conclude that any diXerences we observed structurally were responsible for the conformation change. The purpose of this section was solely to compare the structures to determine if we could find a structural basis for the diXerence between *E. coli* and *C. thermocellum* conformation – we stated a few times throughout the section and in the discussion that there were no immediate structural reasons for this diXerence in shape. We have added a few sentences in the discussion to address whether Gram-positive vs. Gram-negative is influencing the shape, addressed in reviewer #3 comment #4.

(8) Line 237: The whole section "Identification..." analyzed the substrate channel by computational analysis. The author should provide experimental evidence that these residues identified are critical for channeling by generating mutants and measuring their activity.

We agree that mutagenesis is the next logical step for these results, however it is outside the scope of work of this paper as this study will not be that straightforward. We have included a sentence in the discussion to indicate our plans for further investigation to the channel that says, “Future mutagenesis studies will be needed to confirm whether the spirosome exists to control the reaction flux in high-reactant conditions.”

**Reviewer #3 (Recommendations For The Authors):**
(1) The capacity of *C. thermocellum* AdhE spirosomes to switch from a natively extended conformation to a compact conformation is not demonstrated in this manuscript, as it is now. Because this would be the first time that Gram-positive bacterial AdhE spirosomes are observed in a compact conformation, the authors should provide a clear demonstration of their existence by presenting reliable and good images of C. thermocellum compact spirosomes.

We have modified Figure 1A to zoom in on one compact and extended spirosome that we have identified from each *C. thermocellum* sample. We have included triangles of the same size and shape to indicate the proximity of a turn of a helix, showing that the identified compact spirosomes have a tighter conformation than extended spirosomes.

(2) The authors should show at least an image of the compact C. thermocellum spirosomes, that they claim to observe in the presence of NADH or in the forward reaction conditions mentioned in Figure 1. The authors have added diXerent reactants to the extended C. thermocellum spirosomes and visualized their conformation by negative stain. An image of each condition tested would be valuable and would nicely complete the distribution of compact versus extended spirosomes presented in Figure 1.

We have created a new supplemental figure with spirosomes circled for all of the experimental conditions for *C. thermocellum* (Supplemental figure 1). We have added a reference to supplemental figure 1 in the text to direct the reader to these images.

(3) The cryoEM classes presented in Figure 8 are not convincing and could correspond to dimers or rosettes of AdhE or to *E. coli* endogenous AdhE. CryoEM classes showing longer compact *C. thermocellum* spirosomes should be shown. The percentage of these compact spirosomes visualized in the micrographs should be added and discussed in the text as it would increase confidence in these findings and confirm that *C. thermocellum* compact spirosomes exist. Heterologous production of *C. thermocellum* AdhE in *E. coli* depleted for its endogenous AdhE would be required to definitively prove that these are compact *C. thermocellum* AdhE spirosomes in the cryoEM.

We included the pictures of the theoretical compact spirosomes, as generated from the 8-mer of *E. coli* AdhE (6AHC) to address the possibility of rosettes. We have now indicated in the text that there were 6.7% of the particles in the compact conformation, which is less than seen by negative stain. We further mentioned that the compact spirosome is less compact than that seen in *E. coli*. We added a sentence to the discussion about the possibility of contaminating *E. coli* spirosomes (though this is very unlikely) in our compact spirosome analysis: “While these compact spirosomes could result from expression in *E. coli*, though this is very unlikely, we also identified compact spirosomes in a native *C. thermocellum* lysate, which would not have similar contamination issues.”

(4) The authors should include and discuss in the text previous findings (among which Laurenceau et al., 2015...) describing the diXerences between Gram-positive and Gram-negative spirosomes. AdhE spirosomes are natively extended in most Gram-positive bacterial species (S. pneumoniae, S. sanguinis or C. diXicile...), and have never been observed in a compact conformation. On the opposite, *E. coli* (and other Gram-negative bacteria) native AdhE spirosomes are compact and are able to switch to an extended conformation in the presence of the cofactors (NAD+, coA, and iron).

We have added the following sentences to the discussion to address this comment: “This could potentially be due to the diXerences between Gram-positive and Gram-negative bacteria. In previous studies, compact spirosomes have only been isolated from Gram-negatives while solely extended spirosomes have been isolated from Gram-positives. Furthermore, while the compact spirosomes can transition to extended in the presence of cofactors, the reverse has not been previously observed with an extended spirosome.”

(5) The authors have spotted some diXerences between the *E. coli* and *C. thermocellum* structures, that they believe could explain the intrinsic capacity of these spirosomes to be natively extended or compact. It would be interesting to confirm this hypothesis by measuring *C. thermocellum* extended AdhE spirosome activity and comparing it to *E. coli* extended spirosomes. The impact of mutations in the regions proposed by the authors to be important in the capacity of *C. thermocellum* AdhE to be extended (especially the GxGxxG motif and the D494 position) would be appreciated to confirm this hypothesis.

We agree that this would be an interesting avenue of research although it is currently outside the scope of this paper. We are looking into experiments that we can perform where we can track both activity and conformation but have not found an ideal experiment at this time.

(6) Many statements and result interpretations are overstated in several parts of the manuscript and would need to be rewritten to balance the absence of clear evidence of *C. thermocellum* compact spirosomes.

We have shown that we have identified compact spirosomes, addressed in multiple comments above. We have adjusted the language of the paper to indicate more uncertainty that will be followed up in future mutagenesis experiments. However, these mutations are not that simple to identify and this research would require a fairly large study that is better suited for a follow up manuscript.

(7) The Figure 7 legend would need to be corrected.

We are unsure as to what needs to be corrected in the figure 7 legend based on this comment.